**Geoscientific
Instrumentation
Methods and
Data Systems**

**Discussions**

# Optical laboratory facilities at the Finnish Meteorological Institute – Arctic Research Centre

K. Lakkala[1], H. Suokanerva[1], J.M. Karhu[1], A. Aarva[2], A. Poikonen[2], T. Karppinen[1], M. Ahponen[1], H.-R. Hannula[1], A. Kontu[1], and E. Kyrö[1]

[1]Finnish Meteorological Institute – Arctic Research Centre, Tähteläntie 62, 99600 Sodankylä, Finland
[2]Finnish Meteorological Institute, Observation Services, Helsinki, Finland

Received: 8 December 2015 – Accepted: 16 December 2015 – Published: 18 January 2016

Correspondence to: K. Lakkala (kaisa.lakkala@fmi.fi)

Published by Copernicus Publications on behalf of the European Geosciences Union.

Discussion Paper | Discussion Paper | Discussion Paper | Discussion Paper

**GID**

doi:10.5194/gi-2015-43

**Optical laboratory at FMI-ARC**

K. Lakkala et al.

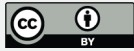

## Abstract

This paper describes the laboratory facilities at the Finnish Meteorological Institute – Arctic Research Centre (FMI-ARC). They comprise an optical laboratory, a facility for biological studies and an office. A dark room has been built, in which an optical table and a fixed lamp test system are set up, and the electronics allow high precision adjustment of the current. The Brewer spectroradiometer, NILU-UV multifilter radiometer and ASD spectroradiometer of the FMI-ARC are regularly calibrated or checked for stability in the laboratory. The facilities are ideal for responding to the needs of international multidisciplinary research, giving the possibility to calibrate and characterize the research instruments as well as handle and store samples.

## 1  Introduction

The location of the Finnish Meteorological Institute Arctic Research Centre (FMI-ARC) (67.367° N, 26.629° E) is ideal for atmospheric and environmental research in the boreal and sub-Arctic zone. Numerous international projects have been conducted in the FMI-ARC, and needs for multidisciplinary laboratory facilities have been obvious. In this paper, we present the optical laboratory facilities of FMI-ARC, and focus on the measurements of optical instruments used for stratospheric and climate research at the FMI-ARC.

The Brewer spectroradiometer (Brewer) measurements (Bais et al., 1993) started in 1988 at Sodankylä. First the focus was in total ozone measurements, as Sodankylä is affected by the spring time Arctic ozone loss in the stratospehre. Since 1990, the Brewer is also used to measure spectral solar ultraviolet (UV) irradiances. The Brewer is a single monochromator with a wavelength range from 290 to 325 nm. The wavelength step of the recorded spectrum is 0.5 nm. The slit function of the Brewer is 0.56 nm at full-width half-maximum (FWHM). In 2012, a second Brewer was purchased to measure next to the old one in the roof of the sounding station of the observatory.

# GID

doi:10.5194/gi-2015-43

**Optical laboratory at FMI-ARC**

K. Lakkala et al.

The second Brewer is a double monochromator with a wavelength range from 290 nm to 365 nm. The Brewer time series is one of the longest homogenised time series measured in the Arctic (Lakkala et al., 2003, 2008).

The longterm solar UV radiation is also measured with multichannel NILU-UV radiometers (Høiskar et al., 2003) located on the roof of the sounding station since 2007. The NILU-UV intruments have also been set up to measure on the peatland field experiment and forest experiment of the Finnish Ultraviolet Internation Research Centre (FUVIRC) during summers of 2002–2011. The NILU-UV monitor the UV-B, UV-A and erythemally-weighted (McKinlay and Diffey, 1987) UV radiation, photosynthetic active radiation and total ozone column, and provides information on cloudiness. The radiometer is a filter instrument with five UV channels, with central wavelengths around 305, 312, 320, 340 and 380 nm, and bandwidths of around 10 nm at FWHM. The sixth channel measures the photosynthetic active radiation (PAR) in the 400–700 nm wavelength region. The radiometer has a Teflon diffuser, silicon detectors, high-quality bandpass filters and is temperature-stabilized to 40 °C. One-minute averages of measured irradiances and detector temperature are recorded.

The analytical Spectral Devices (ASD) Field Spec Pro JR spectroradiometer measures solar UV spectrum in the wavelength region from 350 to 2500 nm covering the longer wavelengths of the UV part of the solar spectrum as well as the visible and near infrared part of the solar spectrum. The measurements at FMI-ARC are used for validation of satellite measurements and algorithm development (e.g., Heinilä et al. (2014); Pulliainen et al. (2014); Niemi et al. (2012)). The measurements started in 2006 and are located on a 30 m height tower (Sukuvaara et al., 2007), from which both reflected radiation of forested and forest opening was measured until 2013. Currently only measurements over the forest are performed.

A common thing for these optical measurements is, that the instruments's measurement capacity tend to change as a function of time. E.g., the sensitivity of the channels of the multifilter radiometer tend to drift over time (Lakkala et al., 2005). In order to obtain reliable and homogenized measurements the instruments need to be well char-

**[GID](doi:10.5194/gi-2015-43)**

doi:10.5194/gi-2015-43

**Optical laboratory at FMI-ARC**

K. Lakkala et al.

Discussion Paper | Discussion Paper | Discussion Paper | Discussion Paper

acterized and regularly calibrated (Webb et al., 1998; Seckmeyer et al., 2001; Webb et al., 2003; Seckmeyer et al., 2010). The quality control and quality assurance of the measurements require monitoring of the stability of the instruments using regular lamp tests, which need to be performed in an appropriate optical laboratory. This work describes the characteristics of the optical laboratory facilities at the FMI-ARC and shows typical measurement protocols for the above mentioned instruments.

## 2  Laboratory facilities

### 2.1  Optical laboratory

The optical laboratory was initially built in 1998, and moved to its present location in November 2002. It is a duplication of the optical laboratory at the observatory of the FMI in Jokioinen. The laboratory comprise two adjacent rooms: the control room and the dark room. The temperature in the rooms is monitored using PT100-sensors, and both rooms are equipped with adjustable air conditioning. The floor is covered with a black plastic membrane and the walls of the dark room are painted with antireflection black paint. Ceiling lamps are also painted black and turned outside of the measurement system. Lockers are covered with antireflection black cloths. An opening has been made on one wall of the dark room to serve as lead-out for cables or installations which need outdoor air.

An optical table, manufacturer Melles Griot, is placed in the the dark room. The average height from the floor is 91.5 cm and its dimesions are 100 × 150 cm. The lamp holders and needed sensors can be fixed to the table with high precision. A lamp holder is set up to fullfill the needs of the calibration of the Brewer spectroradiometer. Also UVB (BN-9102-147 UVB XB03) and UVA (BN-9102-130 UVA XB05) sensors are set up in order to monitor the calibration lamp. The sensors are temperature stabilized using circulating water.

The electronics of the laboratory includes a 0.1Ω shunt resistor (Burster-1282-0.1), a high precision digital multimeter (Hewlett Packard 3458A) and a high precision power supply (Hewlett Packard 6675 A system DC Power supply 0–12 V/0–18 A). A current-to-voltage converter is used to monitor the voltage passing through the lamp. The system allows the current accuracy to be ± 0.001 A. The mentioned electronics are located in the control room, and only the calibration lamp, the dark room temperature sensor and the UVB/A sensors are located in the dark room. The shunt and the multimeter are sent each year to SGS FIMKO Testing and Certification Services, Finland, for calibration.

The circuit diagram of the laboratory is shown in Figure 1. The current can be controlled with the control room PC. For this purpose a LabVIEW System Design Software (LabVIEW), National Instruments, has been tailored to read the voltage drop of the shunt measured by the multimeter, and to regulate the current to the defined value. The final tuning can be done using the adjustable voltage reference connected to the power supply. The readings from temperature and UVB/A sensors and the converter are transferred via a datalogger QLI50 to the control PC.

## 2.2 Laboratory for biological studies

The laboratory for biological studies is located in the next room of the optical laboratory. The room is equipped with machines needed, e.g., for snow and vegetation studies: A temperature chamber, a cold chamber, an ice cube maker, a fume hood and a liquid nitrogen chamber.

The temperature in the temperature chamber (UT12, Thermo) can be regulated to be the ambient temperature $T + 20\,°C \pm 250\,°C$. The regulation range of the cold chamber (SRC 1812/3.1 B (L), Porkka) is $+2\,°C \pm 12\,°C$, and its dimensions are $1800 \times 1200 \times 2000$ mm. The ice cube maker makes 21 kg of ice per 24 h, and it can store 4 kg of ice. The dimension of the liquid nitrogen chamber is 35 liters, the static working time is 130 d, the working time is 80 d and the evaporation rate is $0.27\,L\,d^{-1}$. The fume hood is manufactured by IS VET.

# GID

doi:10.5194/gi-2015-43

**Optical laboratory at FMI-ARC**

K. Lakkala et al.

Discussion Paper | Discussion Paper | Discussion Paper | Discussion Paper

Discussion Paper | Discussion Paper | Discussion Paper | Discussion Paper | Discussion Paper

**GID**

doi:10.5194/gi-2015-43

**Optical laboratory at FMI-ARC**

K. Lakkala et al.

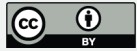

## 3  Measurement procedures

### 3.1  UV spectroradiometer calibrations

The response of the Brewer spectroradiometer is determined by performing 1000 W lamp measurements in the laboratory. The lamps are 1000 W tungsten-filament incandescent halogen lamps of type DXW operated in vertical orientation. The bulbs have been installed in their sockets by Gigahertz Optik. A primary standard is used to transfer the calibration from the National Standard Laboratory MIKES-Aalto. Using the measurements of the Brewer, the irradiance scale is transfered to working standards, which are used for the calibration of the Brewer every six weeks. The procedure is described in more detail in (Mäkelä et al., 2015) in this issue.

Before a calibration the Brewer is moved inside the dark room usually the day before. Also the multimeter is switched on then, which allows both the Brewer and the multimeter to stabilize around 15 h before a calibration. The Brewer is placed on a trolley, which can be fixed and levelled in exactly the same place each time. The Brewer and the lamp were aligned under the same vertical optical axis by using an alignment jig and laser. The distance between the diffuser of the Brewer and the lamp is set to 50 cm. Two baffles are used between the lamp and the Brewer in order to reduce the effect of stray light. The measurement arrangements are shown in Figure 2.

During a lamp measurement, the current is set to 8 A, with an accuracy of ± 0.001 A. The current is controlled by the LabVIEW program, which allows the current to increase slowly during X minutes before reaching the final level. Respectively the current is going down slowly, after finishing the measurements. The lamps typically need 15-20 minutes to stabilize before a measurement can start. The measurement itself takes around 17 minutes, when scanning up to 365 nm. After the measurement, the lamp is left untouched until it has cooled down near room temperature. The temperature of the control and dark rooms is set to 23 ˚C. The ventilation is on during the warming of the lamps, but turned off during the measurements in order to avoid air flows around the lamp. The current, voltage and room temperatures are recorded in a separate metadata

file for each measurement. Also the intensity of the lamp is recorded using the UVB and UVA sensors of the laboratory, so that sudden changes can be noticed.

## 3.2 Stability of the multifilter UV instruments

The fixed set up of the optical laboratory is also used for performing the stability checks of the NILU-UV multichannel radiometers of the FMI-ARC. As routine procedure, the stability of the channels of the NILU-UV is checked twice a year: in spring and in autumn. 100 W OSRAM Radium lamps are mounted in lamp units produced by the manufacturer of the NILU-UV instrument. The lamp unit is connected to the circuit in the place of the lamp (Fig. 1). At least five lamps are used in order to detect the drift of the lamp from the drift of the instrument.

The lamp is warmed up during 5 min in the dark box of the lamp unit at the side of the radiometer. If the lamp would be warmed up at its measurement position on the top of the diffuser, the warming of the radiometer would affect the measurements (Lakkala et al., 2005). The current is increased slowly and set to 6 A by the operator. After the warming, the dark box including the lamp is placed above the diffuser, giving a vertical beam exactly to the same point of the diffuser during each measurement (Fig. 3). The data is recorded with a time step of one second during around 20 s, after which the dark box is removed from the diffuser and the lamp is left to cool down back to the room temperature.

## 3.3 Stability of the UV-VIS spectrometer

The fixed set up of the optical laboratory is also used for monitoring the stabiliity of the ASD field spectroradiometer and the Spectralon reference plate. The Spectralon reference plate is used as a reference for reflection measurements. The lamp measurements are performed once a year with similar 1000 W tungsten-filament incandescent halogen lamps of type DXW operated in vertical orientation as used for the calibra-

## GID

doi:10.5194/gi-2015-43

**Optical laboratory at FMI-ARC**

K. Lakkala et al.

Discussion Paper | Discussion Paper | Discussion Paper | Discussion Paper |

tion of the Brewer spectroradiometers (see Chapter 3.1). The calibration of the lamp is traceable to the National Standard Laboratory MIKES-Aalto.

Before the measurements, the ASD spectroradiometer is left to stabilize at least half an hour in the room temperature. To monitor the stability of the ASD spectroradiometer, and to separate changes in the Spectralon plate from the changes in the spectroradiometer in the reflectance measurement, the spectrum of the lamp is measured with the Remote Cosine Receptor (RCR). The RCR is fixed on the optical table at a vertical distance of 49 cm from the lamp (Fig. 4). One baffle is used to avoid stray light. Reflectance measurements of the Spectralon plates are used for monitoring of the changes in the field Spectralon plate and in the ASD Spectroradiometer itself. Since the field Spectralon is subject to e.g., dust, dirt, snow, rain, freezing, and mechanical stress from continuous use, its properties change. Both the field Spectralon plate and a reference Spectralon plate stored in laboratory conditions are measured. In the reflectance measurements the fibre optic cable is attached to a pistol grip fixed to the table, and the Spectralon plate is set in a fixed position. This setup allows to point the fibre optic cable to exactly the same spot on the Spectralon plates each time. The alignment is adjusted so that the radiation from the lamp is reflected from a Spectralon plate to the fibre optic cable (Fig. 5). The distance between the plate and the lamp is set to 65 cm.

## 4   Conclusions

The optical laboratory facilities at the FMI-ARC comprise a control room, a dark room, a facility for biological studies and an office. They are ideal for calibration and characterization of optical instruments such as spectroradiometers, broadband and multichannel radiometers and aurora cameras. The facilities promote the possiblities for multidisciplinary research. Several international groups have performed studies at FMI-ARC and used the facilities for stratospheric, snow, vegetation and ionospheric studies during the Lapland Atmosphere – Biosphere Facility (LAPBIAT) project under the IHP Access to Research Infrastructures of the European Union. The facilities have served as a

# GID

doi:10.5194/gi-2015-43

## Optical laboratory at FMI-ARC

K. Lakkala et al.

Discussion Paper | Discussion Paper | Discussion Paper | Discussion Paper

central research infrastructure of the Finnish Ultraviolet International Research Center (FUVIRC), where biologists could properly handle and store their samples.

In this work, we presented the set up for calibration of the Brewer spectroradiometers and measuring the stability of the ASD apectroradiometer and the NILU-UV multichannel radiometers of FMI-ARC. The facilities have also been used for characterizations of the instruments, e.g., the temperature and cosine response characterization of the Brewer spectroradiometer have been done in the dark room. Also the aurora cameras of the FMI's network are calibrated in the dark room. The stability of the traveling reference instrument of the NILU-UV Antarctic network was measured once a year in the optical laboratory (Lakkala et al., 2005).

In the dark room, a fixed set up is made for vertical optical axis 1000 W DXW lamp measurements and the electronics allow precise regulation of the current. The optical table is large enough for customized set up for different optical instruments and the rooms have enough space for temporary instruments in order to welcome research groups with different needs.

*Acknowledgements.* Tapani Koskela is acknowledged for the original design of the optical laboratory.

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

# GID

doi:10.5194/gi-2015-43

## Optical laboratory at FMI-ARC

K. Lakkala et al.

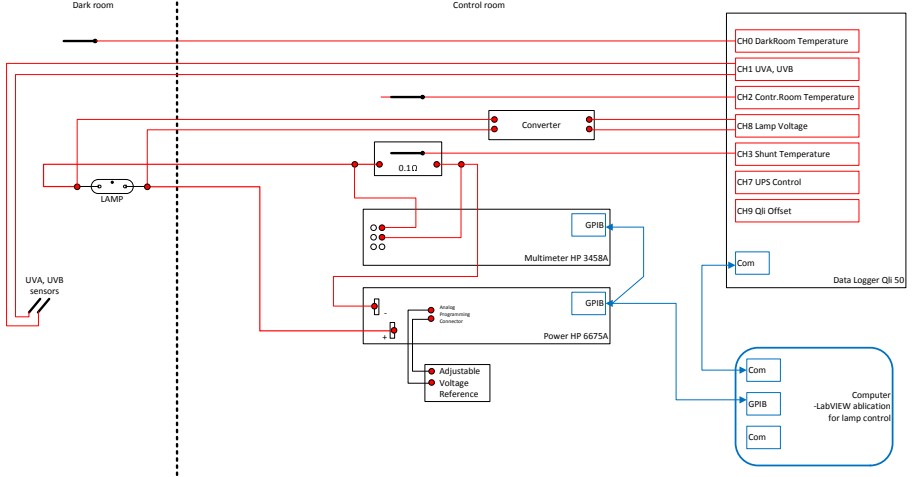

**Figure 1.** The circuit of the optical laboratory in FMI-ARC.



**GID**

doi:10.5194/gi-2015-43

**Optical laboratory at FMI-ARC**

K. Lakkala et al.

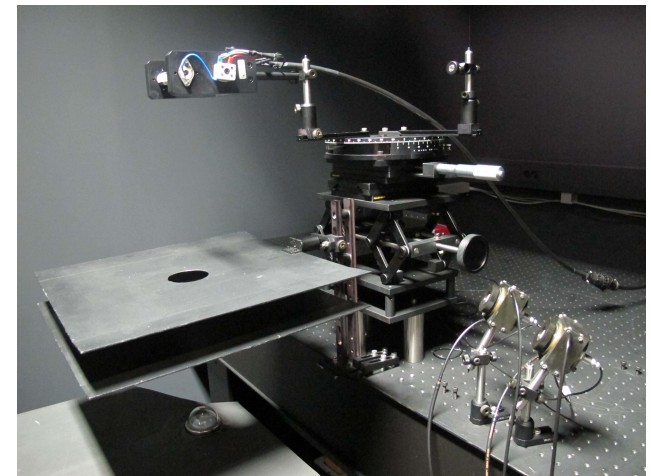

**Figure 2.** Lamp measurement with the Brewer spectroradiometer.

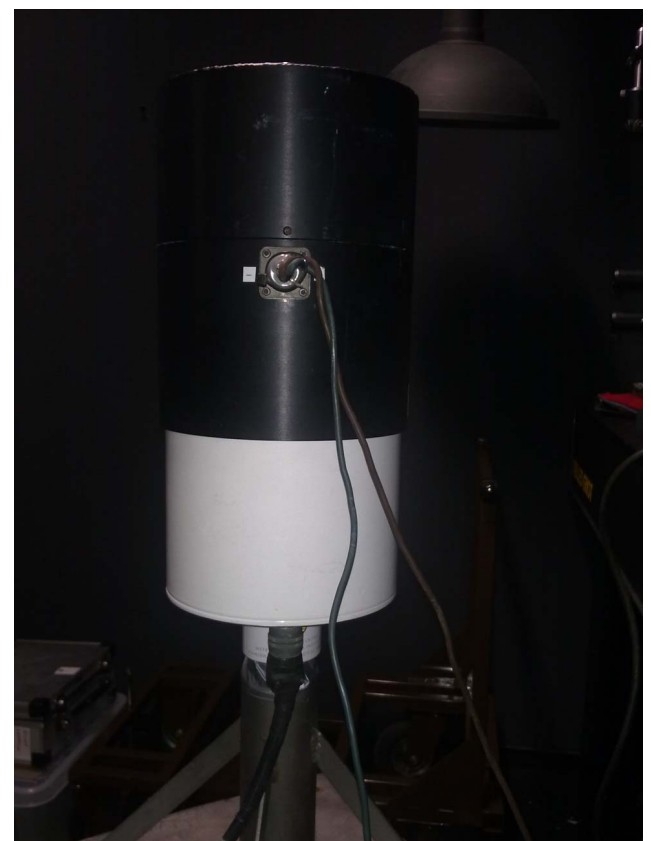

**Figure 3.** Lamp measurement of the NILU-UV radiometer.

**GID**

doi:10.5194/gi-2015-43

**Optical laboratory at FMI-ARC**

K. Lakkala et al.

# GID

doi:10.5194/gi-2015-43

**Optical laboratory at FMI-ARC**

K. Lakkala et al.

Interactive Discussion

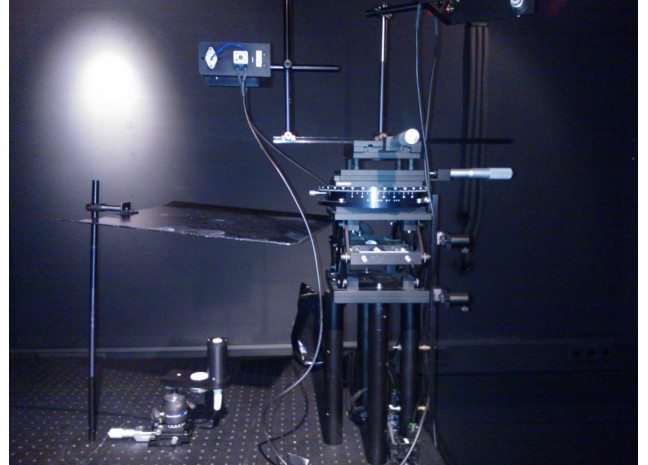

**Figure 4.** The RCR measurement with the ASD FieldSpec Pro JR spectroradiometer.

Discussion Paper | Discussion Paper | Discussion Paper | Discussion Paper

# GID

doi:10.5194/gi-2015-43

## Optical laboratory at FMI-ARC

K. Lakkala et al.

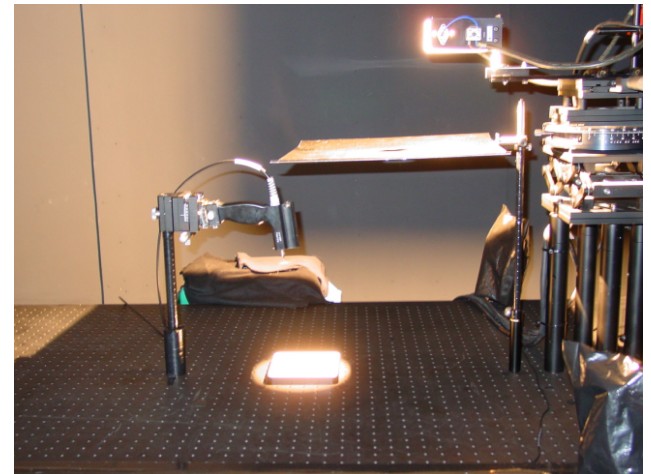

**Figure 5.** The reflectance measurement of the Spectralon reference plate with the ASD Field-Spec Pro Jr spectroradiometer.