# Peer review of "Optical laboratory facilities at the Finnish Meteorological Institute – Arctic Research Centre"

_Geoscientific Instrumentation, Methods and Data Systems, 2015_

## Referee Comment (RC1) · Anonymous Referee #1 · 14 Mar 2016

A) General comments

The manuscript is a clear description of the FMI calibration facility at Sodankylä, Finnland. It explains in full details dark room labor and the different available setups. The clear technical content of the descriptions are most valuable as guideline for similar laboratories and act as reference for scientific work performed at this artic research center.

B) Specific scientific comments

All technical parts of the different calibration and measurement procedures are well described; however the paragraph of the control circuit of the irradiance standard lamp is not easy to follow and needs to be rewritten to improve the quality of the manuscript:

[Figure]

The control of the current passing through the lamp is the most important step for absolute irradiance calibrations. The traceability of this value is usually obtained through a calibrated resistor and a calibrated voltages measuring device (DMM). This should be clearly explained in the corresponding paragraph.

At FMI Sodankylä, the lamp current is acquired by the voltages measurement (HP3458A ) over the reference shunt (Burster 1282, 0.1 Ohm). The details of the calibration of these devices (frequency, uncertainty) could be given. Using thie current value the software controls the lamp current via the power supply (HP6675A). The current uncertainty is finally the combination of the reference devices uncertainties (Shunt and DMM) and the feedback precision.

The reason for the additional devices is unclear: What is the current-to-voltages converter? Looking at the schematic (Fig. 1) it is a looks more like a simple voltage measurement performed from the Data Logger QLI50. This value is a result of the current regulation and thus interesting as a long term stability and quality criterion for the lamp in use.

Finally: What is the fine tuning procedure and what is its purpose?

C.) Presentation

The manuscript is clearly structured. Minor modifications are recommended to improve the quality of the paper:

Page 3, line 7: (Typo) Internation -> International

Page 3, line 17: Reference to the manufacturer ADS Inc.

Page 3, line 4: (Typo) "current passes through the lamp" whereas "Voltages is measured over the lamp". The HP6675A specifications are 0-120V and not 0-12V.

Page 5, line 15: Reference to the manufacturer of the QLI50. I could not find this instrument.

Page 5, line 22: (Typo) T +20 C±250 C -> Probable +-2.5 degC ?

Page 6, line 21: Define "X minutes"

Page7, line 21: (Typo) "the Spectralon . . ." -> "a Spectralon. . ."

Page 8, line 26: IHP = "Improving Human Research Potential", Reference missing

Captions on the Figures are to short; more information is needed. The illumination of the photographs could be much improved.

Figure 1: . . . the electrical circuit. . .

Figure 2: For users who have never seen a brewer irradiance calibrations: The lamp inside the lamp holder is not visible. The Brewer is the white box with the quartz entrance dome seen in the bottom left corner. Indicat the locations of the different devices(lamp in the holder, Brewer, baffle, alignment assembly, Monitor diodes. . ..)

Figure 3: NILU - bottom white, Lamp unit – black top. . .

Figure 4: What is the light spot on the wall? Where is the entrance of the cosine receiver?

Figure 5: Is there a baffle between the spectroradiometer entrance and the reflectance plate? Unlucky angle of the photography.

---

## Referee Comment (RC2) · Anonymous Referee #2 · 31 Mar 2016

Currently the scientific research is more and more frequently oriented on multidisciplinary studies. Local and international projects are conducted. The needs for multidisciplinary laboratory facilities have increased. FMI-ARC at Sodankyla located in sub-Arctic zone is a very good site for environmental and atmospheric research. There is a long-term experience of atmospheric and solar radiation measurements. The measurements with single monochromator based Brewer spectrophotometer were started in 1988 and since then the instrumental and supporting facilities have developed to the current high scientific level. The major content of paper is related to the description of the optical laboratory facilities and the measurement procedures. It is very necessary for the partners in multidisciplinary research. Not always the partners working in other research areas are familiar with the technical facilities of host the hosting institution an there may arise problems during the initial stages of project work. All solar radiation measurements, especially in the UV spectral range, are quite complicated and need applying regular calibration and stability checking measures. It is necessary that the facilities of partners and the hosting institution are in agreement. The published descriptions help to avoid possible complications and at the same time are useful in preparing papers on the results of research. The survey of laboratory facilities and measurements procedures presented in paper is relevant and users friendly. All necessary points are discussed in details. Everybody can get a relevant picture. The language is good enough. Word "also" is used rather too frequently.

---

## Author Comment (AC1) · 20 May 2016

Authors' response to the review of the manuscript "Optical laboratory facilities at the Finnish Meteorological Institute - Arctic Research Centre" by Lakkala et al.

The Authors appreciate the constructive comments of the Referee #1 and respond here below to each remark. The answers are marked as A:. The manuscript has been upgraded following the review comments and uploaded as suplement.

B) Specific scientific comments All technical parts of the different calibration and measurement procedures are well described; however the paragraph of the control circuit of the irradiance standard lamp is not easy to follow and needs to be rewritten to improve the quality of the manuscript: The control of the current passing through the lamp is the most important step for ab- solute irradiance calibrations. The traceability of this

value is usually obtained through a calibrated resistor and a calibrated voltages measuring device (DMM). This should be clearly explained in the corresponding paragraph.

A: The paragraphs describing the control circuit has been modified following the suggestions. They are now :

"For absolute irradiance calibrations, the most important is controlling the current passing through the lamp. The lamp current is acquired by the voltage measurement over the reference shunt and controlled with the control room PC. The electrical circuit diagram of a calibration is shown in Fig. 1, where the main circuit is bolded. The system allows the current accuracy to be $\pm 0.001$A. To ensure the accuracy, the multimeter and the shunt are sent every one-two years to SGS FIMKO Testing and Certification Services, Finland, for checking and calibration. During the last calibration (6.-7.5.2014) the uncertainty of the shunt was $\pm 0.007\%$ which makes $\pm 0.007$m$\Omega$. The reported uncertainty was based on a standard uncertainty multiplied by a coverage factor k = 2, providing a level of confidence of approximately 95%. The last calibration confirmed the specifications of the multimeter provided by the manufacturer, in which the accuracy of the voltage measurement over 24 h was 0.06 ppm.

A LabVIEW System Design Software (LabVIEW), National Instruments, has been tailored to read the voltage drop of the shunt measured by the multimeter, and to regulate the lamp current to the defined value. The regulation can be done in steps of 12 mA using the power supply (HP6675A). As there is a need for higher precision, an adjustable voltage reference with control voltage of 0 - 14 mV is connected to the external analog input of the power supply to finely tune the current regulation (Fig. 1).

In order to safely monitor the voltage over the lamp, which results from the current regulation, a voltage limiter and a voltage converter are need to give the right input signal to the datalogger. Galvanic isolation is made at the same time with the conversion. The voltage readings can be used to monitor the long term stability of the lamp and for quality control of the measurements. The readings from the voltage converter,

temperature and UVB/A sensors are transferred via the datalogger to the control PC."

At FMI Sodankylä, the lamp current is acquired by the voltages measurement (HP3458A ) over the reference shunt (Burster 1282, 0.1 Ohm). The details of the calibration of these devices (frequency, uncertainty) could be given. Using thie current value the software controls the lamp current via the power supply (HP6675A). The current uncertainty is finally the combination of the reference devices uncertainties (Shunt and DMM) and the feedback precision.

A: The uncertainty of the shunt and the accuracy of the multimeter are now given. The text is now:

"The system allows the current accuracy to be $\pm0.001$A. To ensure the accuracy, the multimeter and the shunt are sent every one-two years to SGS FIMKO Testing and Certification Services, Finland, for checking and calibration. During the last calibration (6.-7.5.2014) the uncertainty of the shunt was $\pm0.007\%$ which makes $\pm0.007$m$\Omega$. The reported uncertainty was based on a standard uncertainty multiplied by a coverage factor k = 2, providing a level of confidence of approximately 95%. The last calibration confirmed the specifications of the multimeter provided by the manufacturer, in which the accuracy of the voltage measurement over 24 h was 0.06 ppm."

The reason for the additional devices is unclear: What is the current-to-voltages converter? Looking at the schematic (Fig. 1) it is a looks more like a simple voltage measurement performed from the Data Logger QLI50. This value is a result of the current regulation and thus interesting as a long term stability and quality criterion for the lamp in use.

A: The current-to-voltage converter includes the voltage limiter and the voltage converter with galvanic isolation. It is used to safely monitor the voltage over the lamp. The Figure 1 has been updated. The text has been updated to "In order to safely monitor the voltage over the lamp, which results from the current regulation, a voltage limiter and a voltage converter are need to give the right input signal to the datalogger.

[Figure]

Galvanic isolation is made at the same time with the conversion. The voltage readings can be used to monitor the long term stability of the lamp and for quality control of the measurements. The readings from the voltage converter, temperature and UVB/A sensors are transferred via the datalogger to the control PC. "

The paragraph describing the electronics of the laboratory has been accordingly updated, including informations of the voltage limiter and the voltage to voltage converter:

"The electronics of the laboratory includes a 0.1 $\Omega$ shunt resistor (Burster-1282-0.1), a high precision digital multimeter (Hewlett Packard 3458A), a high precision power supply (Hewlett Packard 6675A system DC Power supply 0-120V/0-18A), a voltage limiter (from 150 VDC to 10 VDC), a voltage to voltage converter with galvanic isolation (Nokeval Signal Converter 641) (from 10 VDC to 2.5 VDC with galvanic isolation), an adjustable voltage reference, a datalogger (QLI50 Sensor Collector, manufactured by Vaisala Oyj) and a control PC. "

Finally: What is the fine tuning procedure and what is its purpose?

A: The following description has been added to the text: "

"A LabVIEW System Design Software (LabVIEW), National Instruments, has been tailored to read the voltage drop of the shunt measured by the multimeter, and to regulate the lamp current to the defined value. The regulation can be done in steps of 12 mA using the power supply (HP6675A). As there is a need for higher precision, an adjustable voltage reference with control voltage of 0 - 14 mV is connected to the external analog input of the power supply to finely tune the current regulation (Fig. 1)."

C.) Presentation The manuscript is clearly structured. Minor modifications are recommended to improve the quality of the paper: Page 3, line 7: (Typo) Internation -> International

A: The text has been corrected. "The NILU-UV instruments have also been set up to measure on the peatland field experiment and forest experiment of the Finnish Ultravi-

olet International Research Centre (FUVIRC) during summers of 2002-2011."

Page 3, line 17: Reference to the manufacturer ADS Inc.

A: The suggested reference has been added. The text is now: "The spectroradiometer of type Analytical Spectral Devices (ASD) FieldSpec Pro JR Full Range, manufactured by Analytical Spectral Devices, Inc., nowadays PANalytical, measures solar UV spectrum in the wavelength region from 350 nm to 2500 nm covering the longer wavelengths of the UV part of the solar spectrum as well as the visible and near infrared part of the solar spectrum."

Page 3, line 4: (Typo) "current passes through the lamp" whereas "Voltages is measured over the lamp". The HP6675A specifications are 0-120V and not 0-12V.

A: The text has been corrected following the remark:

"...a high precision power supply (Hewlett Packard 6675A system DC Power supply 0-120V/0-18A)".

And the typo has been corrected, as the corresponding sentence is now: "In order to monitor the voltage over the lamp which results from the current regulation, a voltage limiter and a voltage converter are used."

Page 5, line 15: Reference to the manufacturer of the QLI50. I could not find this Instrument.

A: As suggested, a reference to the manufacturer has been added: "The readings from temperature and UVB/A sensors and the converter are transferred via a datalogger QLI50 Sensor Collector, manufactured by Vaisala Oyj, to the control PC."

Page 5, line 22: (Typo) T +20 C 250 C -> Probable +-2.5 degC ?

A: The text has been corrected: " The temperature in the temperature chamber (UT12, Thermo) can be regulated between ambient temperature T (+20degC) and +250degC. The regulation range of the cold chamber (SRC 1812/3.1 B (L), Porkka) is between

+2degC and +12degC, and its dimensions are 1800 mm x 1200 mm x 2000 mm. "

Page 6, line 21: Define "X minutes"

A: The text has been updated: "The current is controlled by the LabVIEW program, which allows the current to increase slowly during 2-3 minutes before reaching the final level."

Page7, line 21: (Typo) "the Spectralon" -> "a Spectralon"

A: the text has been corrected following the suggestion: "The fixed setup of the optical laboratory is also used for monitoring the stability of the ASD field spectroradiometer and a Spectralon reference plate."

Page 8, line 26: IHP = "Improving Human Research Potential", Reference missing

A: The text has been updated following the suggestion. "Several international groups have performed studies at FMI-ARC and used the facilities for stratospheric, snow, vegetation and ionospheric studies during the Lapland Atmosphere - Biosphere Facility (LAPBIAT) project under the Improving Human Research Potential – Access to Research Infrastructures of the European Union (contract no. 025969-TA, www.sgo.fi/lapbiat)."

Captions on the Figures are to short; more information is needed. The illumination of the photographs could be much improved.

A: Unfortunately the authors didn't find better photographs, as it is hard to take pictures of black instruments in black surroundings. However the authors think that they are good enough to give an idea of the measurement installation. More information has been added to the caption of the Figures, see below.

Figure 1:the electrical circuit

A: The caption of Figure 1 has been corrected following the suggestion: "The electrical circuit of the optical laboratory in FMI-ARC. The main circuit is bolded. The vertical

black dashed line denotes the separation between the dark room and the control room. Instruments at its left side are in the dark room, while the instruments in the right side are in the control room."

Figure 2: For users who have never seen a brewer irradiance calibrations: The lamp inside the lamp holder is not visible. The Brewer is the white box with the quartz entrance dome seen in the bottom left corner. Indicat the locations of the different devices(lamp in the holder, Brewer, baffle, alignment assembly, Monitor diodes.)

A: More information has been added to the Figure caption:"Lamp measurement with the Brewer spectroradiometer. The diffuser of the Brewer is on the left bottom corner under the baffles. The lamp is placed in the lamp holder over the baffles. The goniometer is seen in the middle and the UVA and UVB sensors on the right bottom corner."

Figure 3: NILU - bottom white, Lamp unit – black top

A: More information has been added to the Figure caption:"Lamp measurement of the NILU-UV radiometer. The NILU-UV is the white cylinder. The lamp is placed inside the black cylinder on the top of the NILU-UV. "

Figure 4: What is the light spot on the wall? Where is the entrance of the cosine Receiver?

A: The light spot on the wall is reflection of the laboratory light, which are off during the measurements.The entrance of the cosine receiver is the small white spot straight under the lamp.

Please also note the supplement to this comment:
http://www.geosci-instrum-method-data-syst-discuss.net/gi-2015-43/gi-2015-43-AC1-supplement.pdf

43, 2016.

[Figure]

[Figure]

**Fig. 1.**

[Figure]

**Fig. 2.**

[Figure]

[Figure]

**Fig. 3.**

[Figure]

**Fig. 4.**

[Figure]

**Fig. 5.**

[Figure]

---

## Author Comment (AC2) · 20 May 2016

Authors' response to the review of the manuscript "Optical laboratory facilities at the Finnish Meteorological Institute - Arctic Research Centre" by Lakkala et al.

The Authors appreciate the comments of the Referee #2 and respond here below to the remark. The manuscript has been updated following the answers to the remark. The updated manuscript is uploaded as supplement.

Comment from the Referee #2: Word "also" is used rather too frequently.

Answer: The word "also" has been avoided in the following sentences:

page 3, row 6: "In addition, the NILU-UV instruments have been set up to measure on the peatland field experiment and forest experiment of the Finnish Ultraviolet International Research Centre (FUVIRC) during summers of 2002-2011."

page 4 rows 14-16: "The floor is covered with a black plastic membrane and the walls together with the ceiling lamps of the dark room are painted with antireflection black paint. Ceiling lamps are turned away from the measurement system."

page 6, row 12: "The multimeter is switched on then, which allows both the Brewer and the multimeter to stabilize around 15 h before a calibration."

page 7, row 1: "The intensity of the lamp is recorded using the UVB and UVA sensors of the laboratory, so that sudden changes can be noticed."

Please also note the supplement to this comment:
http://www.geosci-instrum-method-data-syst-discuss.net/gi-2015-43/gi-2015-43-AC2-supplement.pdf
* * *
[Figure]

**Supplement:**

Manuscript prepared for J. Name
with version 2015/09/17 7.94 Copernicus papers of the LaTeX class copernicus.cls.
Date: 20 May 2016

[revised manuscript text omitted]

The electronics of the laboratory includes a 0.1 $\Omega$ shunt resistor (Burster-1282-0.1), a high precision digital multimeter (Hewlett Packard 3458A), a high precision power supply (Hewlett Packard 6675A system DC Power supply 0-120V/0-18A), a voltage limiter (from 150 VDC to 10 VDC), a voltage to voltage converter with galvanic isolation (Nokeval Signal Converter 641) (from 10 VDC to 2.5 VDC with galvanic isolation), an adjustable voltage reference, a datalogger (QLI50 Sensor Collector, manufactured by Vaisala Oyj) and a control PC. The mentioned electronics are located in the control room, and only the calibration lamp, the dark room temperature sensor and the UVB/A sensors are located in the dark room.

For absolute irradiance calibrations, the most important is controlling the current passing through the lamp. The lamp current is acquired by the voltage measurement over the reference shunt and controlled with the control room PC. The electrical circuit diagram of a calibration is shown in Fig. 1, where the main circuit is bolded. The system allows the current accuracy to be $\pm0.001$A. To ensure the accuracy, the multimeter and the shunt are sent every one-two years to SGS FIMKO Testing and Certification Services, Finland, for checking and calibration. During the last calibration (6.-7.5.2014) the uncertainty of the shunt was $\pm0.007\%$ which makes $\pm0.007$ m$\Omega$. The reported uncertainty was based on a standard uncertainty multiplied by a coverage factor k = 2, providing a level of confidence of approximately 95%. The last calibration also confirmed the specifications of the multimeter provided by the manufacturer, in which the accuracy of the voltage measurement over 24 h was 0.06 ppm.

A LabVIEW System Design Software (LabVIEW), National Instruments, has been tailored to read the voltage drop of the shunt measured by the multimeter, and to regulate the lamp current to the defined value. The regulation can be done in steps of 12 mA using the power supply (HP6675A). As there is a need for higher precision, an adjustable voltage reference with control voltage of 0 - 14 mV is connected to the external analog input of the power supply to finely tune the current regulation (Fig. 1).

In order to safely monitor the voltage over the lamp, which results from the current regulation, a voltage limiter and a voltage converter are need to give the right input signal to the datalogger. Galvanic isolation is made at the same time with the conversion. The voltage readings can be used to monitor the long term stability of the lamp and for quality control of the measurements. The readings from the voltage converter, temperature and UVB/A sensors are transferred via the datalogger 
[revised manuscript text omitted]

Lakkala, K., Jaros, A., Aurela, M., Tuovinen, J.-P., Kivi, R., Suokanerva, H., Karhu, J., and Laurila, T.: Radiation measurements at the Pallas-Sodankylä Global Atmosphere Watch station — diurnal and seasonal cycles of ultraviolet, global and photosynthetically-active radiation, Boreal Env. Res., 21, 427–444, 2016.

Mäkelä, J., Lakkala, K., Meinander, O., Kaurola, J., Koskela, T., Karhu, J. M., Karppinen, T., de Leeuw, G., and Heikkilä, A.: In search of traceability: Two decades of calibrated Brewer UV measurements in Sodankylä and Jokioinen, Geosci. Instrum. Method. Data Syst. Discuss., in review, doi:10.5194/gi-2015-40, 2016.

McKinlay, A. F. and Diffey, B. L.: A reference action spectrum for ultraviolet induced erythema in human skin, CIE J., 6, 17–22, 1987.

Niemi, K., Metsämäki, S., Pulliainen, J., Suokanerva, H., Böttcher, K., Leppäranta, M., and Pellikka, P.: The behaviour of mast-borne spectra in a snow-covered boreal forest, Remote Sensing of Environment, 124, 551–563, doi:10.1016/j.rse.2012.06.008, 2012.

Pulliainen, J., Salminen, M., Heinilä, K., Cohen, J., and Hannula, H.-R.: Semi-empirical modeling of the scene reflectance of snow-covered boreal forest: validation with airborne spectrometer and lidar observations, Remote Sens. Environ., 155, 303–311, 2014.

Seckmeyer, G., Bais, A., Bernhard, G., Blumthaler, M., Booth, C., Disterhoft, P., Eriksen, P., McKenzie, R., Miyauchi, M., and Roy, C.: Instruments to Measure Solar Ultraviolet Radiation,Part 1: Spectral Instruments, World Meteorological Organization (WMO), Global Atmosphere Watch Report No. 125, 2001.

Seckmeyer, S., Bais, A., Bernhard, G., Blumthaler, M., Johnsen, B., Lantz, K., and McKenzie, R.: Instruments to Measure Solar Ultraviolet Radiation, Part 3: Multi-channel filter instruments, World Meteorological Organization (WMO), Global Atmosphere Watch Report No. 190, 2010.

Sukuvaara, T., Pulliainen, J., Kyrö, E., Suokanerva, H., Heikkinen, P., and Suomalainen, J.: Reflectance spectroradiometer measurement system in 30 meter mast for validating satellite images, IGARSS: 2007 IEEE International Geoscience and Remote Sensing Symposium, pp. 2885–2889, 2007.

240     Webb, A., Gardiner, B., Martin, T., Leszcynski, K., Metzdorf, J., and Mohnen, V.: Guidelines for Site Quality
        Control of UV Monitoring, World Meteorological Organization (WMO), Global Atmosphere Watch Report
        No. 126, 1998.

        Webb, A., Gardiner, B., Leszczynski, K., Mohnen, V., Johnston, P., Harrison, N., and Bigelow, D.: Quality As-
        surance in Monitoring Solar Ultraviolet Radiation: the State of the Art, World Meteorological Organization
245     (WMO), Global Atmosphere Watch Report No. 146, 2003.

[Figure]

**Figure 1.** The electrical circuit of the optical laboratory in FMI-ARC. The main circuit is bolded. The vertical black dashed line denotes the separation between the dark room and the control room. Instruments at its left side are in the dark room, while the instruments in the right side are in the control room.

[Figure]

**Figure 2.** Lamp measurement with the Brewer spectroradiometer. The diffuser of the Brewer is on the left bottom corner under the baffles. The lamp is placed in the lamp holder over the baffles. The goniometer is seen in the middle and the UVA and UVB sensors on the right bottom corner.

[Figure]

**Figure 3.** Lamp measurement of the NILU-UV radiometer. The NILU-UV is the white cylinder. The lamp is placed inside the black cylinder on the top of the NILU-UV.

[Figure]

**Figure 4.** The Remote Cosine Reseptor measurement with the ASD FieldSpec Pro JR spectroradiometer.The entrance of the Cosine Reseptor is the small white spot on the top of the black cylinder straight under the lamp. The light spot on the wall is reflection from the laboratory light, which is off during the measurements.

[Figure]

**Figure 5.** The reflectance measurement of the Spectralon reference plate with the ASD FieldSpec Pro Jr spectroradiometer. The reference plate in placed on the table and the fibre optic cable is attached to the pistol grip fixed to the table in order to measure the reflected lamp radiation. The lamp is attached to the holder in the right top corner.